# [Re] Panoptic-DeepLab: A Simple, Strong, and Fast Baseline for Bottom-Up Panoptic Segmentation

## 1 Reproducibility Summary

### 2 Scope of Reproducibility

3 The original work by Cheng et al. [5] introduces Panoptic-DeepLab - a novel architecture for panoptic segmentation,
4 claiming to achieve comparable performance to two-stage, top down approaches while yielding fast inference speeds.
5 At the time of publication, Panoptic-Deeplab claims to have ranked first in all three cityscapes benchmarks *(specifically:*
6 *mIoU, AP & PQ).*

### 7 Methodology

8 As the original paper authors published their source code, our codebase integrates sections of their codebase, while
9 re-implementing components intrinsic to the main claim we are attempting to evaluate. We also studied the source code,
10 using information provided from it and the pipelines to augment our understanding from what the paper described.

11 While we initially attempted a code-blind reproduction, it was soon determined to be unfeasible following which a
12 hybrid approach was instantiated.

### 13 Results

14 While we successfully reproduced the given architecture, we have been unable to train it. Therefore: our contributions
15 currently remaining exclusive to architecture, and certain unit tests within the system itself. We also highlight potential
16 low-level Tensorflow that were pitfalls to our development, that may be advantageous to investigate.

### 17 What was easy

18 The authors of the paper structured their contributions on well-documented and tested frameworks such as ResNet and
19 DeepLabV3+, while training on popular datasets such as Cityscapes and Mapillary Vistas. Consequently, setting up the
20 dataset and the environment to reproduce the given research was straightforward.

### 21 What was difficult

22 A significant hurdle we came across during our reading of the paper was vagueness within the expected implementation.
23 This extended from the architecture to the training regime. The descriptions provided, although accurate, were presented
24 as a high-level overview, with the expectation of a lot of prior domain knowledge. This resulted in a significant time-sink,
25 following which we looked into the codebase for necessitated context.

26 Despite the well-structured objected oriented implementation through which the code was written, we found certain
27 sections hard to understand. We observed convoluted re-implementations of high level functions already part of
28 Tensorflow as part of the codebase. However, this could have been a direct result of the implementation not using the
29 now-popularised Functional API within Tensorflow, which may have resulted in the required use of custom layers.

## Communication with original authors

We communicated with the authors over e-mail, resolving doubts that arose while reading the paper. It is also through author communication that we were directed to the codebase, as although public - the relevant repository wasn't mentioned within the paper.

# 1   Introduction

Since it's inception in 2018, Panoptic Segmentation[10] has remained a popular task within the domain of computer vision. It is in effect the unification of two distinct yet related tasks, namely: semantic and instance segmentation. Semantic segmentation broadly involves the assignment of a class ID to every input pixel, whereas instance segmentation is the delineation of distinct objects within an input frame. Broadly classified as "stuff" and "things", the unification of the two produces the target output known as Panoptic Segmentation.

Panoptic-Deeplab[5] aims to establish a strong baseline for a bottom-up approach to the task. Consequently, it places a focus on simplicity, cleverly incorporating established components within neural architecture to set state-of-art benchmarks as of the date of publication.

# 2   Scope of reproducibility

We investigate the following claims from the original paper:

- Panoptic-DeepLab establishes a solid baseline for bottom-up methods that can achieve comparable performance of two-stage methods while yielding fast inference speed - nearly real-time on the MobileNetV3 backbone.

- Single Panoptic-DeepLab simultaneously ranks first (at the time of publication) at all three Cityscapes benchmarks, setting the new state-of-art of $84.2\%$ mIoU, $39.0\%$ AP, and $65.5\%$ PQ on test set.

# 3   Methodology

Initially, we attempted a code-blind reproduction of Panoptic-DeepLab. However, we swiftly determined it to be unfeasible - primarily as a result of us being unable to fully grasp implementation details from the paper itself. The paper does incredibly well to provide a high level explanation of how the architecture functions; unfortunately, the lack of implementation-specific information prevented a blind-paper reproduction without extensive interpolation. Crucially: we note the importance of a standardized system for presenting architecture diagram. While the current abstract layers look nicer, we find they lack important information necessary to reproduction.

It is important to note here that upon re-reading the paper post implementation - with a prior understanding of the architecture - we found that just the paper did very well to explain the architecture, enough even, for a code-blind reproduction. Going through long-expired threads of discussion was an exercise that did well to remind us of implicit interpolations we made, having already known the architecture.

## 3.1   Model description

Panoptic-DeepLab[5] incorporates an encoder-decoder architecture to generate target inference, with our implementation encapsulating $6,547,894$ total parameters, of which $6,534,998$ are trainable, while the remaining $12,896$ are non-trainable. Broadly - it sequentially incorporates the modules discussed in the following subsections.

### 3.1.1   Image DataGenerator

To the extent of our understanding, Panoptic-Deeplab[5] did not discuss the implementation of its dataset loaders. As a result, we entirely used a custom implementation of Tensorflow's ImageDataGenerator[1] class, to function as an iterator for the training regime. Since we did not find it highlighted within the paper to generate ground-truth center heatmaps and centerpoint predictions, we discuss this in the following paragraph.

**Center Heatmaps & Prediction[13]**   The center heatmaps & prediction maps are representations of the ground truth instance ID images. These images are effective data representations of instances within the frame. Each 'thing' has an encoded value, for instance: each pixel representing car#1 may be labeled 10001, while car#2 is labelled 10002. The first two digits encode one of the 19 different objects tracked by Cityscapes - in this case, the car - while the final three digits refer to the instance of the given object. The representation in specific are the computed averages of each of the instances - producing the center prediction. The center heatmaps are a gaussian distribution applied over the centerpoint predictions with $standard\ deviation = 8px$.

### 3.1.2 Encoder

Panoptic-DeepLab is trained on three popular encoder ImageNet pre-trained backbones, namely: Xception-71[6], ResNet-50[8] & MobileNetV3[9]. The backbone works to generate feature maps from input images. For the purpose of this reproduction, we use Xception-71 as our encoder backbone, as this is the primary implementation used by the original authors. We integrate our own implementation of the Xception-71 module as part of the paper reproduction.

### 3.1.3 Atrous Spatial Pyramid Pooling

From the encoder, the feature maps are split into dual modules. The first layer to run the decoupled modules is Atrous Spatial Pyramid Pooling[4], abbreviated - ASPP, is a module that concurrently resamples encoded feature layers at different rates, finally pooled together to capture objects and relevant context at multiple scales.

We derived the ASPP block directly from the tensorflow implementation maintained by the paper authors, with no modifications made to the architecture.

### 3.1.4 Decoder

Panoptic-DeepLab is a fork of the DeepLabV3+[4] decoder architecture. It incorporates two fundamental contributions, specifically: an additional skip connection in the upsampling stage, and an additional upsampling layer with `output stride = 8`. We developed a custom implementation of this utilizing the modern Keras Functional[2] API. Through our development of the decoder, we ran into a prominent problem, that delayed significantly our progress within model architecture. This is in direct correlation with how Tensorflow handles internal API calls, type conversion.

**tf.Tensor v KerasTensor**    `KerasTensor` is an internal class within the Keras API. It is generated during layer definition, during the construction of a neural architecture. When latent features are passed during the function calls, the `KerasTensor` object is converted implicitly to the `tf.Tensor` format - covering up significant type discrepancies. As part of testing the original Panoptic-Deeplab code, we evaluated that as part of the model conversion to the Functional API, it was unable to retrace inputs to the decoder. This resulted in a Graph Disconnected error. In an attempt to allow traceback to work, we devised an approach wherein skip connections were made instance variables within the Decoder class, and passed separately to the functional call. It is here that we discovered that the lack of the implicit type conversion, while transferring precisely the same set of data resulted in a TypeError. We were unable to manually make the necessary conversion, highlighting a lack of documentation as `KerasTensor` is a backend class. Consequently, we were unable to patch the approach and proceeded to a full rewrite.

**Graph Disconnected**    An error we struggled to get past - the Graph Disconnected error is thrown when the traceback method within the functional API is unable to generate the necessary I/O graph to create a valid architecture. While in retrospect: the information provided was enough to debug effectively the point of failure, we would like to highlight that we believe a more visual or verbose representation - for instance, a plot describing the graph upto the point of failure - may allow the quicker & clearer identification of the issue.

### 3.1.5 Prediction Heads

The decoupled decoder modules further split into three separate prediction heads. These generate the final deep-learning based output within our implementation. They are a final set of convolutional followed by fully connected layers generating the final result.

Similar to ASPP[4], we derived prediction heads directly from the tensorflow implementation maintained by the paper authors, with no modifications made to the architecture.

### 3.1.6 Loss Function

Panoptic-DeepLab employs a collective loss function intended to train resultant outputs.

$$L = \lambda_{sem}L_{sem} + \lambda_{heatmap}L_{heatmap} + \lambda_{offset}L_{offset}$$

This was a straightforward function, the implementation of which was just as straightforward, and did not require any effort above the requisite minimum.

### 3.1.7 Post Processing

Post processing of the outputs heads in effect involves stitching the instance and semantic segmentation outputs via a majority vote, generating the final panoptic segmentation. Since output post processing involves a traditional script with no trainable parameters, we have used post-processing code directly from the original tensorflow implementation, as put forward by the authors of the paper.

### 3.2 Datasets

Panoptic-DeepLab used Cityscapes[7], Mapillary Vistas[12] & COCO[11] datasets over the proposed architecture. For the purpose of our implementation, we train our model on the Cityscapes dataset, as examples are referenced from it through the evaluation stages of the model. Each image is of size $(1025, 2049)$, and utilizes an odd crop size to allow centering, aligning features across spatial resolutions.

### 3.3 Hyperparameters

Panoptic-DeepLab uses a training protocol similar to that of the original DeepLab, specifically: the 'poly' learning rate policy. It uses the `Adam` optimizer with a learning rate of $0.001$ without weight decay, with fine-tuned batch normalization parameters and random scale data augmentation. While we prepared our re-implementation with the same set of hyperparameters, we were unable to validate our approach, further discussed in Section 3.5.

### 3.4 Experimental setup and code

Alongside git for code tracking, we also employ data science specific tools such as DVC (Data Version Control) and MLFlow[3] with DAGsHub as the platform operating the relevant stack of services. DVC requires S3 buckets, that maintain the dataset, models, visualization and high storage binaries utilized during training. MLFlow - specifically, MLFlow tracking was the service we utilized as part of documenting the training lifecycle, including experimentation, and the relevant comparison between training cycles.

### 3.5 Computational requirements

By an astronomical margin, the computational requirements necessary for training Panoptic-DeepLab was the factor that prevented us from successfully testing our target reproduction. Originally, the architecture was trained on a cluster of *32 TPUs*. In a technical report that detailed a PyTorch re-implementation of Panoptic-DeepLab, they coupled runtime optimization techniques alongside smaller batch size to reduce the training size to *4-8 GPUs*. While a significant improvement, we find that stating it enables 'everyone be able to reproduce state-of-the-art results with limited resources' a vast extension.

The computational stack under active access to our team includes a single GPU on a docker container, personal workstations as well as any GPUs provisioned by cloud notebook service *Google Colaboratory*. Even considering the use of cloud compute services such as *AWS* - that are estimated to cost upwards of $2,000$ USD - for the acquisition of necessary compute, it is not possible to acquire access to the high performance GPU-enabled G3 instances without explicit approval from AWS customer support. Through a back-and-forth that extended across weeks, we have been unable to acquire the approval necessary to create stated instances.

We therefore attempted the utilization of CPU resources to train the model to the best of our ability. We theorized the use of high learning rates in an attempt to overfit the model in a single epoch as a sanity check; to ensure the pipeline for our re-implementation worked as intended. Predictably, the training failed, and python was killed as the memory usage exceeded the cap permitted by the system, causing it to crash.

## 4   Results

As a result of the scenario detailed in the previous section: while we did manage to reproduce the architecture, we have been - as of now - unable to train it. Therefore, to this degree, our reproduction has not been a success, with our contributions currently remaining exclusive to architecture and the challenges encountered by us through our reproduction of the paper.

# 5 Discussion

Through the constant cycle of updates across which the languages on which neural architectures are written, the Reproducibility Challenge presents the fantastic opportunity to (1) take a step back, and (2) re-approach a pre-existing codebases with an entirely different perspective. It allows us the opportunity to fine-tune both past research and research in the near future. The insights our team has generated from our work on Panoptic-DeepLab itself, has done immensely to broaden our own perspective on the state of our field at the moment.

## 5.1 What was easy

The authors of the paper structured their contributions on well-documented frameworks such as ResNet and DeepLabV3+, while training on popular datasets such as Cityscapes and Mapillary Vistas. Consequently, setting up the dataset and the environment to reproduce the given research was straightforward.

Additionally, various modules within the architecture were concisely and concretely defined - which enabled us to re-implement them without additional effort, above the minimum requisite. We found several sections of the paper were written with meticulous detail, and we especially appreciated the exhaustive, vast array of experiments and benchmarks provided as part of the research, which led our primary motivations towards attempting the reproduction.

## 5.2 What was difficult

A significant hurdle we came across during our reading of the paper was vagueness within the expected implementation. This extended from the architecture to the training regime. The descriptions provided, although accurate, were presented as a high-level overview, with the expectation of a lot of prior domain knowledge. This resulted in a significant time-sink, following which we looked into the codebase for necessitated context.

Despite the well-structured objected oriented implementation through which the code was written, we found certain sections hard to understand. We observed convoluted re-implementations of high level functions already part of Tensorflow as part of the codebase. However, this could have been a direct result of the implementation not using the now-popularised Functional API within Tensorflow, which may have resulted in the required use of custom layers.

Additionally, we would also like to highlight the importance of excessive computational requirements within the machine learning space, and it's relation to the reproducibility of a paper. With the exploding costs of GPUs owing to extensive crypto-mining farms[14], and the ever increasing complexity of models being trained over time, it is imperative to consider designing systems that adhere to development policies ranging beyond the best-funded labs, and represents an important milestone within the democratization of research within high-throughput deep learning.

## 5.3 Communication with original authors

We enjoyed minimal yet significant communication with the original authors of the research. We communicated over e-mail, resolving doubts we came across as we read the paper. We found valuable insight through this communication, which has consequently been imperative to the success of our project. It has enabled discovering an additional suite of supplementary literature written with respect to the target architecture, which we may have potentially been unable to find without significant delay.

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
