# OpenReview forum: "[Re] Panoptic-DeepLab: A Simple, Strong, and Fast Baseline for Bottom-Up Panoptic Segmentation"
_ML_Reproducibility_Challenge/2021/Fall — Reject_

### Official Review · Reviewer_r6tk · 2022-03-01
**Good start, but incomplete**

**Rating:** 4
**Confidence:** 3

**Review:**

The authors of this submission took on a particularly complex and expensive reproduction task. They describe interesting lessons along the way, but did not produce empirical results.

A lot of the ideas and lessons along the way are really good if they were fully fleshed out. For instance, early on the submission points out that there is not "standardized system for presenting architecture diagram[s]." It's also been my experience that these diagrams are usually drawn in a way that roughly borrows techniques and forms from similar papers, but based on subjective interpretations rather than a standard and precise system. Referring to architecture diagrams throughout reading is often really crucial to understanding papers like Panoptic-DeepLab, so this is an important shortcoming of the literature. A submission fully describing a good formal procedure to produce these diagrams would be incredibly interesting and useful, but this submission does not pursue it further.

The authors also list difficulties with Tensorflow development that echo those that many researchers describe when explaining why Tensorflow is not the preferred framework for most published DL research. This is a document that I'd hope that the Tensorflow team also reads, but it is not going to be suprising to most of its users. Github issues are probably the best venue to raise them. I'm uncertain, though, whether the authors are describing a bug in the original authors' implementation, or unclear error messages and poor design in Tensorflow encountered during the submission's initial attempt at a code-blind reproduction. I'd appreciate this being clarified.

The paper also says that they encountered "vagueness within the expected implementation" and that there was "the expectation of a lot of prior domain knowledge." It'd be helpful to have details in both cases. Where the original authors are vague, it'd be a useful contribution for this submission to point out specific passages and how they found the correct interpretation. Prior domain knowledge that isn't properly described and cited would also be very important to point out, but I'm unclear what specific knowledge was missing.

The submission also touches on the topic of how the cost and resources used by experiments is related to reproducibility. It presents a useful example of the tradeoffs. But I'd want to emphasize that the submission is talking about reproducibility *and* "democratization" of research in section 5.2, but these are very different issues that should not be conflated too much. If work cannot be reproduced by another lab with similar resources as the original authors, this is a matter of scientific validity and correctness. If a work cannot be reproduced because labs have dissimilar levels of access to the necessary resources, this is a matter of policy and how science is funded.

I'd summarize the result as: the authors took on a paper that requires far more software engineering time and compute resources to implement than most AI papers. The authors of the reproduction do not have these resources, so they were not able to so. While the observations along the way are interesting, this is less valuable than a complete reproduction. So I'd rate it lower than papers that do produce empirical results, and overall recommend rejection.

---

### Official Review · Reviewer_C71M · 2022-03-16
**Reproducing Panoptic-DeepLab**

**Rating:** 3
**Confidence:** 4

**Review:**

The authors discuss the limitations they faced while reproducing the Panoptic-DeepLab paper.
1. Their major limitation was access to computing resources for training Panoptic-DeepLab. Panoptic-Deeplab requires a minimum of 4-8 GPUs for training the networks. It is still surprising that the authors were unable to get access to AWS compute even after weeks. The authors then reverted to using CPUs for training the model. As a reviewer, I would like to see if the authors tried to train the Panoptic-DeepLab network on a single GPU. A simple way would have been to use a simpler backbone like AlexNet, use lower-resolution images, use mixed-precision training, gradient accumulation over multiple iterations, etc. to utilize the single GPU available to them to the full extent. I do not see this in the report.
2. The authors discuss software limitations that are better suited for a GitHub issue rather than a report on reproducing Panoptic-DeepLab.

---

### Meta-Review · Area_Chair_zHEV · 2022-04-08

**Recommendation:** Reject
**Confidence:** 5

**Metareview:**

Although the authors provide some useful insights, the reproducibility study is not complete.

---

### Decision · Program_Chairs · 2022-04-09

Reject